# Prognostic Impact of Low Skeletal Muscle Mass on Major Adverse Cardiovascular Events in Coronary Artery Disease: A Propensity Score-Matched Analysis of a Single Center All-Comer Cohort

**DOI:** 10.3390/jcm8050712

**Published:** 2019-05-19

**Authors:** Dong Oh Kang, So Yeon Park, Byoung Geol Choi, Jin Oh Na, Cheol Ung Choi, Eung Ju Kim, Seung-Woon Rha, Chang Gyu Park, Suk-Joo Hong, Hong Seog Seo

**Affiliations:** 1Cardiovascular Center, Division of Cardiology, Department of Internal Medicine, Korea University Guro Hospital, Korea University College of Medicine, Seoul 08308, Korea; gelly9@naver.com (D.O.K.); trv940@naver.com (B.G.C.); koolup93@gmail.com (J.O.N.); wmagpie@korea.ac.kr (C.U.C.); withnoel@hanmail.net (E.J.K.); swrha617@yahoo.co.kr (S.-W.R.); parkcg@kumc.or.kr (C.G.P.); 2Department of Radiology, Korea University Guro Hospital, Korea University College of Medicine, Seoul 08308, Korea; iasy1124@naver.com (S.Y.P.); hongsj@korea.ac.kr (S.-J.H.)

**Keywords:** cardiovascular disease, coronary artery disease, computed tomography, L1 SMI, percutaneous coronary intervention, sarcopenia, skeletal muscle index

## Abstract

The impact of sarcopenia on atherosclerotic cardiovascular disease remains unclear. We aimed to investigate the prognostic impact of sarcopenia on coronary artery disease (CAD). A total of 475 patients with CAD who underwent successful percutaneous coronary intervention (PCI) and computed tomography (CT) were enrolled. The cross-sectional area of skeletal muscle at the first lumbar (L1) vertebral level was measured, and sex-specific cut-off values of L1 skeletal muscle index (L1 SMI; male <31.00 cm^2^/m^2^, female <25.00 cm^2^/m^2^) were obtained. The primary outcome was 3-year all-cause mortality and the secondary outcome was 3-year major adverse cardiovascular events (MACEs). Low L1 SMI was present in 141 (29.7%) of 475 patients. The incidence of all-cause mortality (23.7% vs. 5.9%, *p * < 0.001) and MACEs (39.6% vs. 11.8%, *p * < 0.001) was significantly higher in patients with low L1 SMI than in those with high L1 SMI. In multivariate analysis, low L1 SMI was an independent predictor of higher risk of all-cause mortality (hazard ratio (HR): 4.07; 95% confidence interval (CI): 1.95–8.45; *p * < 0.001) and MACEs (HR: 3.76; 95% CI: 2.27–6.23; *p * < 0.001). These findings remained consistent after propensity score-matched analysis with 91 patient pairs (C-statistic = 0.848). CT-diagnosed low skeletal muscle mass is a powerful predictor of adverse outcomes in patients with CAD undergoing PCI.

## 1. Introduction

Sarcopenia is characterized by age-related generalized loss of skeletal muscle mass (SMM) and muscle strength, and frequently accompanies metabolic, physical, and functional disabilities [1,2,3,4]. According to recent consensus recommendations, sarcopenia can be defined as reduced muscle mass, with values less than two standard deviations below the mean value in a healthy young adult population [3,4,5].

Previous studies have reported sarcopenia as a potential predictor of the prevalence and prognosis of atherosclerotic cardiovascular disease (ASCVD) [6,7,8,9,10,11,12,13,14,15,16,17]. Most of those previous studies have used traditional anthropologic measurements [6], bioelectrical impedance analysis (BIA) [6,7,8], or dual X-ray absorptiometry (DEXA) [9,10] to measure SMM. Although these methods can provide estimates of skeletal muscle mass, computed tomography (CT) has been proposed as the gold-standard method for muscle mass measurement [3,4]. Previous studies using CT to measure muscle mass in patients with cardiovascular disease were mostly confined to those undergoing transcatheter aortic valve implantation [11,12], endovascular aneurysm repair for abdominal aortic aneurysm [13,14,15], and those with peripheral artery disease [16,17]. To date, no studies have used CT to determine the presence and long-term prognostic impact of low SMM in patients with coronary artery disease (CAD) who underwent successful coronary stenting.

This study investigated the long-term prognostic impact of CT-diagnosed low SMM in patients with CAD who underwent successful percutaneous coronary intervention (PCI). We identified the sex-specific cut-off values of SMM at the first lumbar (L1) vertebral level to predict all-cause mortality and validated its prognostic significance.

## 2. Material and Methods

### 2.1. Study Population

The PCI registry of Korea University Guro Hospital (KUGH; Seoul, Republic of Korea) is a single-center, prospective, all-comer cohort that consecutively enrolls patients undergoing PCI for CAD. The study population was enrolled from the KUGH PCI registry between January 2004 and April 2014, and was analyzed retrospectively. The study protocol complied with the Declaration of Helsinki and was approved by the Ethics Committee and Institutional Review Board at KUGH (2018GR0352). From the KUGH PCI registry, 788 patients who underwent PCI for CAD with stent implantation and performed CT scans within 180 days of PCI were selected. The following were excluded sequentially: (1) Patients without successful PCI; (2) patients who did not undergo CT within 30 days of PCI; and (3) CT exams without cross-sectional images at the L1 level. Finally, a total of 475 patients were included in the study (Appendix A).

### 2.2. Patient Management, Data Collection, and Clinical Follow-up

All participants in the present study underwent successful PCI. PCI was considered successful if the final angiography showed residual stenosis of <30% with achievement of Thrombolysis in Myocardial Infarction grade 3 flow. The morphology of the coronary artery lesion type was determined according to the American College of Cardiology and American Heart Association classification [18]. Multivessel disease was defined as >50% diameter stenosis of two or more major coronary arteries. Post-PCI medications, including dual anti-platelet therapy, beta-blockers, calcium channel blockers, renin–angiotensin system blockers, and statins, were prescribed according to contemporary practice guidelines.

At initial admission for PCI, demographic features and underlying cardiovascular risk factors were recorded through detailed patient interviews, and baseline laboratory results including lipid profiles were obtained. The Canadian Study of Health and Aging (CSHA) Clinical Frailty Scale (CFS) [19] was used to measure the level of frailty prior to admission. The presence of frailty was defined by CSHA CFS ≥5, with mild to moderate (CFS 5–6) and severe (CFS 7) frailty classifications. The left ventricular ejection fraction (LVEF) was determined with transthoracic echocardiography using a modified Simpson’s biplane method. CT was obtained within 30 days of PCI, and the study was classified as either elective or emergent for acute symptoms. CT protocols included the following five categories: Chest CT, coronary CT, abdomen–pelvis CT, lower extremity CT angiography, and torso positron emission tomography-CT. CT scans with cross-sectional images at the L1 level were analyzed.

After discharge, a regular outpatient clinic visit was scheduled at the end of the first month and every 3 to 6 months thereafter. If a patient did not show up for a scheduled visit, a telephone interview was conducted as an alternative approach for assessment of adverse outcomes. The follow-up duration was commenced with the date of successful PCI.

### 2.3. Study Outcomes and Definition

The primary outcome was 3-year all-cause mortality, and the secondary outcome was 3-year major adverse cardiovascular events (MACEs), a composite of all-cause mortality, non-fatal myocardial infarction (MI), and repeat revascularization. Mortality was classified as cardiovascular or non-cardiovascular. MI was defined as a significant elevation of cardiac biomarkers accompanied by concomitant symptoms or electrocardiographic findings indicative of ischemia [20]. Repeat revascularization included clinically driven revascularization by either PCI or bypass surgery after discharge from the index procedure.

### 2.4. Definition of Low SMM and Method of SMM Assessment

CT measurement of the skeletal muscle area (SMA) of a specific body part accurately reflects the muscle mass of that region [21], and the SMA measured at the third lumbar (L3) vertebral level is widely accepted as a reliable estimate of whole-body SMM [22,23]. Most previous studies on sarcopenia used SMA at the L3 level as a reference. In patients with CAD, CT is usually focused on the thoracic and upper lumbar vertebral levels. Therefore, cross-sectional images at the L3 level are often not available. In such circumstances, cross-sectional images from other lumbar vertebral levels can be used as reliable alternatives [22,24]. In the present study, the L1 level was selected as a reference due to greater availability on both chest and coronary CT scans.

Transverse CT images obtained at the level of the L1 transverse process were assessed in each patient. The CT Hounsfield unit ranges of −29 to 150 and −190 to –30 were considered to represent skeletal muscle and adipose tissue, respectively. The total SMA at the L1 level was composed of the following four components: Paraspinal muscle, extracostal abdominal wall muscle (including external and internal oblique abdominal muscle, latissimus dorsi, and rectus abdominis muscle), intercostal muscle (including transversus abdominis muscle), and psoas muscle and diaphragm. Cross-sectional areas (cm^2^) of skeletal muscles on CT images were measured manually by utilizing the freehand module of the PACS workstation (G3 Infinitt PACS; Infinitt Healthcare, Seoul, Republic of Korea) by two independent observers (one radiologist and one cardiologist). Both observers were blinded to the clinical information during the measurement period. Repeat measurements were performed at least 4 weeks after the initial measurement to assess intra- and inter-observer variability with 50 randomly selected CT images. The cross-sectional areas were normalized for height (cm^2^/m^2^) and presented as the L1 skeletal muscle index (L1 SMI). Due to the absence of established reference values for defining sarcopenia with the L1 SMI in an Asian population, sex-specific cut-off point analysis was performed with the L1 SMI to determine the best reference value for predicting the primary outcome episodes. Based on the determined reference values, patients were divided into two groups by the presence of low SMM. Figure 1 shows representative CT images of one patient with low L1 SMI and one with high L1 SMI.

### 2.5. Statistical Analysis

Data are expressed as mean ± standard deviation (SD) or median with interquartile range (IQR) for continuous variables, and frequency (percent) for categorical variables. Differences between groups were analyzed using Student’s *t*-test or the Mann–Whitney U-test for continuous variables, and Pearson’s chi-square or Fisher’s exact test for categorical variables. Sex-specific reference values for the L1 SMI were derived from the Youden index of the receiver operating characteristic (ROC) curve. The cumulative incidence rate in each group was analyzed using the Kaplan–Meier method with the log-rank test. Multiple stepwise Cox-proportional hazards regression analysis was conducted to test the prognostic impact of low L1 SMI and frailty, and four models were established using subsequent adjustment of confounding factors. To adjust for additional confounding factors, propensity score-matched analysis was performed using a logistic regression model. The following factors, which showed significant differences at baseline or had proper relevance, were included in the model: Age, sex, body mass index (BMI), presentation as MI, hypertension, diabetes, dyslipidemia, previous malignancy, clopidogrel and statins at discharge, multivessel disease, creatinine clearance <60 mL/min, LVEF <50%, total cholesterol, number of treated lesions and treated vessels, number of implanted stents, average stent diameter, total stent length, and implantation of second generation drug-eluting stents. The logistic model estimating the propensity score revealed a good predictive value (C-statistic = 0.848). Intra- and inter-observer variability was quantified using the intra-class correlation coefficient (ICC) and visualized with Bland–Altman plots. Significant differences were defined at a p-value of <0.05, and all analyses were two-tailed. All statistical analyses were performed using Statistical Package for the Social Sciences (SPSS) software, version 20.0 (SPSS-PC Inc., Chicago, IL, USA).

## 3. Results

### 3.1. CT Measurement of L1 SMI and Reference Values for Defining Low SMM

The average duration from the date of PCI to CT scan was −3.3 ± 11.7 days (median −2.0 days). Appendix A shows the sex-specific ROC curves generated to predict both primary and secondary outcomes from the L1 SMI. The areas under the curves (AUCs) for 3-year all-cause mortality in male and female subjects were 0.771 (95% confidence interval (CI): 0.685 to 0.857; *p * < 0.001; Appendix A) and 0.643 (95% CI: 0.510 to 0.776; *p* = 0.045; Appendix A), respectively. The optimal sex-specific cut-off value of L1 SMI for defining low SMM was 31.00 cm^2^/m^2^ (sensitivity 73.60%, specificity 67.70%) in males and 25.00 cm^2^/m^2^ (sensitivity 76.20%, specificity 52.60%) in females.

Based on these reference values, low L1 SMI was present in 141 (29.7%) of the 475 study patients. The measured L1 SMI values of the low L1 SMI and high L1 SMI groups were 26.52 ± 3.74 and 38.36 ± 5.24 cm^2^/m^2^ in males, respectively, and 21.77 ± 2.64 and 31.36 ± 4.02 cm^2^/m^2^ in females, respectively. The L1 SMI showed a weak correlation with BMI (Appendix A). Both intra- and inter-observer variability for the L1 SMI were considered excellent, with optimal ICC values of >0.75 (Appendix A). Details of the CT scans are provided in Appendix A.

### 3.2. Baseline Characteristics and Procedural Profiles

Mean follow-up duration was 4.11 ± 3.02 years (median 4.04 years), and 82.3% of the study participants completed a 3-year follow-up. Baseline clinical characteristics, procedural characteristics, and treatment strategies are displayed in Table 1. The low L1 SMI group was older, and had a lower BMI and greater level of frailty than the high L1 SMI group. Laboratory results including lipid profiles, creatinine clearance, and LVEF showed significantly lower values in the low L1 SMI group than in the high L1 SMI group. The low L1 SMI group showed lower prescription rates for clopidogrel and statins after PCI. Otherwise, the proportion of underlying medical risk factors and most of the procedural profiles were comparable in both groups. 

### 3.3. Clinical Outcomes According to the Presence of Low L1 SMI

A comparison of 3-year clinical outcomes between the groups is displayed in Table 2 and Figure 2. Compared to the high L1 SMI group, the low L1 SMI group had significantly higher incidence of 3-year all-cause mortality (23.7% vs. 5.9%; hazard ratio (HR): 4.34; 95% CI: 2.45 to 7.69; *p* < 0.001) and MACEs (39.6% vs. 11.8%; HR: 3.82; 95% CI: 2.54 to 5.74; *p * < 0.001). Other patient-oriented outcome measures such as MI and repeat revascularization also showed significantly higher incidence in the low L1 SMI group than in the high L1 SMI group. Detailed information about the cause of non-cardiovascular death is provided in Appendix A.

### 3.4. Multivariate Analysis for the Prognostic Impact of Low L1 SMI

Table 3 shows the results of multiple stepwise Cox regression analysis using sequential adjustment of potential confounding factors. The presence of low L1 SMI was an independent predictor of both 3-year all-cause mortality and MACEs after adjustment for age, sex, BMI (model 2), and factors showing a significant difference at baseline (model 3). In the fully adjusted model (model 4), low L1 SMI remained as a major contributing factor for the development of 3-year all-cause mortality and MACEs (HR: 4.07; 95% CI: 1.95 to 8.45; *p * < 0.001 and HR: 3.76; 95% CI: 2.27 to 6.23; *p * < 0.001, respectively). The complete dataset for univariate analysis and stepwise multivariate analysis of the study population is displayed in Appendix A.

### 3.5. Propensity Score-Matched Analysis

Propensity score-matched analysis yielded 91 pairs of study participants. The baseline characteristics and procedural profiles were well balanced between the two groups (Table 1). Similar to the results for the whole study population, the incidence of 3-year all-cause mortality and MACEs was significantly higher in the low L1 SMI group than in the high L1 SMI group (25.7% vs. 10.0%, *p* = 0.009; and 46.1% vs. 18.1%, *p * < 0.001, respectively) (Table 2). Low L1 SMI remained as a strong risk factor for both 3-year all-cause mortality and MACEs (HR: 2.68; 95% CI: 1.23 to 5.83; *p* = 0.013; and HR 2.80; 95% CI: 1.56 to 5.03; *p* = 0.001, respectively) (Table 3).

### 3.6. Clinical Outcomes Based on the Presence of Frailty

Detailed study results based on the presence of frailty are provided in Appendix A. Frail patients showed significantly higher incidence of 3-year all-cause mortality (24.8% vs. 6.4%; HR: 3.99; 95% CI: 2.28 to 6.98; *p * < 0.001) and MACEs (33.4% vs. 17.8%; HR: 2.07; 95% CI: 1.37 to 3.13; *p * < 0.001) than non-frail patients. In fully adjusted multivariate analysis, the presence of frailty was an independent predictor of 3-year all-cause mortality (HR: 2.81; 95% CI: 1.47 to 5.39; *p* = 0.002). However, the impact of frailty on 3-year MACEs was markedly attenuated (HR: 1.29; 95% CI: 0.81 to 2.05; *p* = 0.28).

### 3.7. Detailed Characteristics of Patients Excluded at Initial Enrollment

A modest proportion (39.7%, *n =* 313) of patients were excluded from the initial enrollment, mostly due to the absence of CT scans within 30 days of PCI and unavailability of SMA data at the L1 level. Details of study participants and those excluded at initial enrollment are displayed in Appendix A and Appendix A. The enrolled study participants tended to have worse risk factor profiles, with presentations as MI, histories of diabetes and peripheral artery disease, greater levels of frailty, and lower levels of LVEF and creatinine clearance with a borderline statistical significance. The excluded patients were more likely to have worse clinical features, with higher levels of BMI and lower implantation rates of drug-eluting stents. Despite these differences at baseline, 3-year clinical outcomes were comparable in both groups, except for a significant difference in non-cardiovascular death.

## 4. Discussion

The presence of low SMM defined by the L1 SMI on CT was strongly associated with increased risk of both 3-year all-cause mortality and MACEs in patients with CAD who had undergone successful PCI. The results from the stepwise multivariate analysis and propensity score-matched population also confirmed that low L1 SMI was an independent predictor of 3-year adverse clinical outcomes.

### 4.1. Novelty of the Present Study

To the best of our knowledge, this is the first real-world study to demonstrate the long-term prognostic impact of CT-diagnosed low SMM using the L1 SMI in patients with established ASCVD, in particular those who have undergone successful PCI for CAD. Diagnosis of low SMM using measurement of L1 SMI is a novel method that has not been frequently used in previous studies of sarcopenia. This study also provided a reference value for L1 SMI that can be used for defining sarcopenia in an Asian population. This finding has strong prognostic implications by predicting all-cause mortality. The reference values in this Asian population were lower than the recently reported diagnostic cut-off values for L1 SMI of <34.60 cm^2^/m^2^ for males and <25.90 cm^2^/m^2^ for females in the North American population [24], reflecting the differences in SMM between races and ethnicities [25].

Only one study in a small sample of patients with non-small cell lung cancer used L1 SMI for the diagnosis of sarcopenia [26]. However, no studies have reported the prognostic significance of low SMM defined by L1 SMI. The cross-sectional images at the L1 level were assessed for the diagnosis of low SMM in the present study, and the prognostic value of L1 SMI was validated in detail in patients with established ASCVD. The measurement of L1 SMI on CT showed highly reproducible results with excellent intra- and inter-observer agreement.

Previous studies reported that sarcopenia assessed with various methods was closely associated with subclinical features of CAD, such as coronary artery calcification [8,27] and subclinical coronary artery stenosis [28]. However, the prognostic significance of sarcopenia in CAD based on real-world data has not been reported. The present study is significant as it provides new evidence linking sarcopenia and poor clinical outcomes in CAD.

### 4.2. Assessment of SMM by L1 SMI as a Promising Marker of Prognosis and Comorbidity

The results of the present study suggest that diagnosis of low SMM within 30 days of PCI using the L1 SMI could be used as a powerful prognostic marker in patients with CAD. The presence of low SMM provides additional prognostic information and complements traditional risk factors. Low L1 SMI was associated with significantly higher risk of both fatal and less severe clinical outcomes, including repeat revascularization. Although the risk of cardiovascular death was attenuated in the propensity score-matched population, a 1.6-fold increased risk of cardiovascular death still remained in the low L1 SMI group. The risk of non-cardiovascular death was significantly increased by more than 4-fold in the low L1 SMI group, suggesting that low SMM is likely to be accompanied by various medical comorbidities during long-term follow-up. Previous studies have reported that sarcopenia is closely related to the presence of medical comorbidities including chronic degenerative diseases [29,30,31]. These higher comorbidities associated with low SMM are expected to have greater impact on non-cardiovascular death than cardiovascular death, and negatively affected the overall clinical outcomes in this study population. The findings of the present study suggest that CAD patients with low SMM should consider undergoing comprehensive investigation to discover accompanying medical comorbidities.

In the present study, patients with low L1 SMI and frailty showed a similar pattern of the risk of 3-year adverse outcomes. The negative impacts of frailty on patients with CAD undergoing PCI were consistent with the previous study results [32]. It is well known that both sarcopenia and frailty share overlapping characteristics in a variety of pathophysiologic conditions [33,34,35]. The CFS was able to predict the risk of 3-year all-cause mortality similarly to the CT-determined low L1 SMI. However, prediction of other patient-oriented outcomes were limited. Although CT scans are expensive and associated with radiation burden, detailed characterization of muscle loss is able to provide better diagnostic and prognostic information. Further studies to investigate the prognostic impact of sarcopenia assessed by alternative methods, such as BIA or DEXA, should be conducted in this population, and comparison with frailty scales should be performed to provide additional evidence.

Although a causal relationship with the clinical outcome of CAD could not be established based on the present results, low SMM could be a potential therapeutic target for reducing adverse clinical outcomes in patients with CAD. Since sarcopenia is affected by multifactorial etiologies, a comprehensive multidisciplinary approach including both pharmacological and non-pharmacological interventions should be considered [36]. Cardiovascular rehabilitation has a Class I recommendation for patients undergoing PCI for CAD, and the beneficial effect of physical rehabilitation has been widely investigated [37,38,39]. As a representative non-pharmacological intervention, physical rehabilitation could prevent further progression of sarcopenia and result in better clinical outcomes.

### 4.3. Mechanisms Linking Sarcopenia to Its Prognostic Value

Previous studies have reported that sarcopenia determined using various methods is closely associated with a higher prevalence of cardiometabolic risk factors and established ASCVD [8,9,28]. The presence of sarcopenia is also a contributing factor in poor cardiopulmonary function in patients with CAD [10]. Although the detailed mechanisms should be further investigated, the higher prevalence of cardiometabolic risk factors and functional disabilities that accompanies sarcopenia and low SMM might have contributed to the worsened clinical outcomes in the present study. Another possible underlying mechanism for this worsened prognosis is the diminished endocrine function of muscle cells in patients with sarcopenia. The muscle cells exert an endocrine function by secreting myokines with beneficial cardiovascular effects [40]. Both a decrease in the amount of muscle cells and a decline in their endocrine function in patients with sarcopenia could have contributed to the poor clinical outcomes.

### 4.4. Limitations and Future Perspectives

First, the CT measurement of L1 SMI was not a primary goal. However, the image quality was optimal in most cases and L1 SMI measurement was highly reproducible. Under the widespread use of coronary CT in CAD, CT images to assess SMM at the L1 level could be easily acquired without additional radiation burden or extra payment in this population. Second, because of the intrinsic limitation of non-randomized registry data, the baseline differences between groups could have affected the outcome. To minimize such biases, multiple stepwise Cox regression and propensity score-matched analyses were performed. However, remaining confounders could have influenced the analyses. Third, a modest proportion (*n =* 313; 39.7%) of patients were excluded from the initial enrollment, mostly due to the absence of CT scans within 30 days of PCI and unavailability of SMA data at the L1 level. For balanced interpretation of the study results, detailed features of the excluded study population were provided and compared to those of the enrolled study participants in the Online Appendix A. The results of the present study should be interpreted by considering the minor differences between the included and excluded populations. Fourth, the sample of 475 patients was relatively large compared to that in previous studies of sarcopenia using CT in research on cardiovascular disease [11,12,13,14,15,16,17,28]. However, this sample size was not large enough to draw solid conclusions. Further studies with a larger study population and multicenter enrollment should be conducted to provide greater statistical power and confirm the reproducibility of the results.

## 5. Conclusions

Low SMM is a powerful predictor of adverse clinical outcomes in patients with CAD who undergo successful PCI. CT-diagnosed low SMM may further aid in risk stratification and decision-making in patients with established ASCVD.

## Figures and Tables

**Figure 1 jcm-08-00712-f001:**
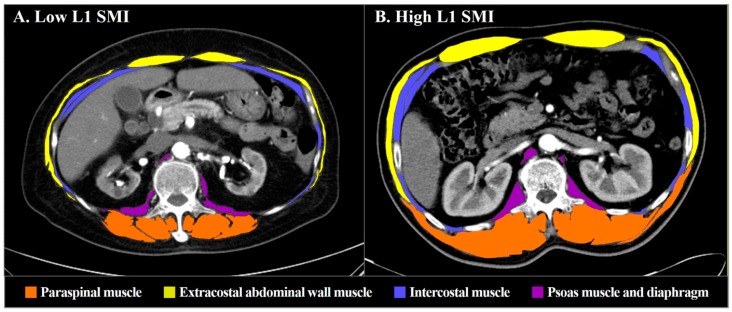
Representative CT images of L1 skeletal muscle area measurement: A patient with low L1 SMI (**A**) and a patient with high L1 SMI (**B**). Total skeletal muscle area was measured using the following four components: Paraspinal muscle (red), extracostal abdominal wall muscle (yellow), intercostal muscle (blue), and psoas muscle and diaphragm (purple). The measured L1 SMAs were 36.24 (**A**) and 149.56 cm^2^ (**B**), and L1 SMIs were 16.00 (**A**) and 49.68 cm^2^/m^2^ (**B**), respectively. CT = computed tomography; L1 = first lumbar vertebra; SMA = skeletal muscle area; SMI = skeletal muscle index.

**Figure 2 jcm-08-00712-f002:**
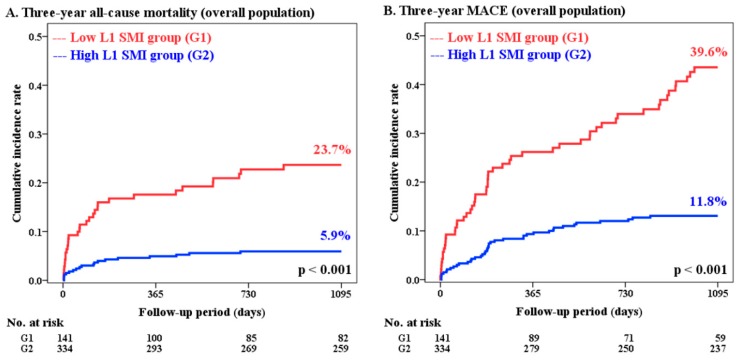
Cumulative incidence rates of 3-year clinical outcomes according to the groups divided by L1 SMI. Results for the overall population (**A**,**B**) and PSM population (**C**,**D**) are displayed: 3-year all-cause mortality (**A**,**C**) and 3-year MACEs (**B**,**D**). L1 = first lumbar vertebra; MACE = major adverse cardiovascular event; PSM = propensity score-matched; SMI = skeletal muscle index.

**Table 1 jcm-08-00712-t001:** Baseline characteristics and procedural profiles.

	Overall Population				PSM Population			
	Low L1 SMI (*n =* 141)	High L1 SMI (*n =* 334)	*p*-Value	S.diff	Low L1 SMI (*n =* 91)	High L1 SMI (*n =* 91)	*p*-Value	S.diff
Age (years)	70.4 ± 11.0	64.0 ± 9.2	<0.001	−0.65	68.7 ± 11.5	67.1 ± 8.7	0.26	−0.17
Sex (men)	100 (70.9)	230 (68.8)	0.66	−0.05	66 (72.5)	65 (71.4)	0.87	−0.03
BMI (kg/m^2^)	21.6 ± 2.6	24.9 ± 2.9	<0.001	1.16	22.8 ± 2.3	23.0 ± 2.6	0.47	0.11
**Clinical presentation**								
Myocardial infarction	46 (32.6)	97 (29.0)	0.44	−0.09	29 (31.8)	28 (30.7)	0.87	−0.03
Unstable angina	36 (25.5)	110 (32.9)	0.11	0.20	24 (26.3)	29 (31.8)	0.42	0.15
Stable angina	48 (34.0)	115 (34.4)	0.94	0.01	30 (32.9)	29 (31.8)	0.87	−0.03
**Past medical history**								
Previous CAD	4 (2.8)	6 (1.7)	0.49	−0.26	2 (2.1)	0 (0.0)	0.50	0.10
Hypertension	85 (60.2)	223 (66.7)	0.18	0.16	59 (64.8)	53 (58.2)	0.36	−0.15
Diabetes	68 (48.2)	145 (43.4)	0.34	−0.11	43 (47.2)	46 (50.5)	0.66	0.07
Dyslipidemia	18 (12.7)	65 (19.4)	0.08	0.28	14 (15.3)	15 (16.4)	0.84	0.05
Cerebrovascular accident	12 (8.5)	28 (8.3)	0.96	−0.01	9 (9.8)	5 (5.4)	0.41	−0.35
Peripheral artery disease	29 (20.5)	51 (15.2)	0.16	−0.20	20 (21.9)	16 (17.5)	0.46	−0.15
Previous malignancy	13 (9.2)	19 (5.6)	0.16	−0.29	6 (6.5)	6 (6.5)	>0.99	<0.001
Current smoker	48 (34.0)	115 (34.4)	0.94	0.01	34 (37.3)	36 (39.5)	0.76	0.05
Frailty (CFS ≥5)	47 (33.3)	74 (22.1)	0.011	−0.31	30 (32.9)	23 (25.2)	0.25	−0.21
Mild to moderate (CFS 5–6)	35 (24.8)	70 (20.9)			21 (23.0)	22 (24.1)		
Severe (CFS 7)	12 (8.5)	4 (1.1)			9 (9.8)	1 (1.0)		
**Laboratory data**								
Total cholesterol (mmol/L)	3.88 (3.18–4.64)	4.20 (3.49–5.12)	0.004	0.13	3.88 (3.41–4.63)	4.03 (3.23–4.71)	0.90	0.01
LDLc (mmol/L)	2.34 (1.77–3.10)	2.64 (1.99–3.36)	0.034	0.10	2.35 (1.89–3.09)	2.37 (1.77–3.00)	0.78	0.02
hs-CRP (mg/L)	7.0 (1.0–17.0)	2.0 (1.0–8.5)	0.006	−0.13	6.0 (1.0–15.5)	4.0 (1.0–17.5)	0.80	−0.02
HbA1c (%)	6.41 ± 1.47	6.56 ± 1.77	0.40	0.09	6.31 ± 1.60	6.61 ± 1.79	0.24	0.18
CrCl (mL/min)	60.4 ± 29.4	79.4 ± 30.6	<0.001	0.63	66.1 ± 30.9	71.3 ± 27.1	0.23	0.18
LVEF (%)	50.6 ± 11.4	54.2 ± 9.6	0.001	0.35	51.2 ± 11.3	51.6 ± 12.9	0.85	0.03
**CT scan profiles**								
Average days from PCI to CT scan	−2.5 ± 11.6	−3.7 ± 11.8	0.30	−0.10	−4.0 ± 10.8	−4.0 ± 10.5	0.96	−0.01
**L1 SMA (cm^2^)**	66.92 ± 15.86	96.49 ± 21.84	<0.001	1.46	70.18 ± 16.06	90.30 ± 17.17	<0.001	1.21
Male	74.07 ± 12.61	107.37 ± 16.22	<0.001	2.18	77.60 ± 11.65	98.43 ± 11.58	<0.001	1.79
Female	49.49 ± 6.90	72.45 ± 10.45	<0.001	2.45	50.60 ± 7.00	69.98 ± 10.82	<0.001	2.13
**L1 SMI (cm^2^/m^2^)**	25.14 ± 4.066	36.18 ± 5.864	<0.001	2.05	26.03 ± 4.03	33.95 ± 4.52	<0.001	1.85
Male	26.52 ± 3.74	38.36 ± 5.24	<0.001	2.39	27.49 ± 3.45	35.54 ± 3.77	<0.001	2.23
Female	21.77 ± 2.64	31.36 ± 4.02	<0.001	2.60	22.19 ± 2.74	29.99 ± 3.81	<0.001	2.35
**Post-PCI medications**								
Aspirin	127 (90.0)	309 (92.5)	0.38	0.17	82 (90.1)	81 (89.0)	0.81	0.05
Clopidogrel	116 (82.2)	301 (90.1)	0.017	0.37	77 (84.6)	76 (83.5)	0.84	−0.05
Beta-blocker	70 (49.6)	159 (47.6)	0.68	−0.05	45 (49.4)	44 (48.3)	0.89	0.05
Calcium channel blocker	49 (34.7)	101 (30.2)	0.33	−0.11	31 (34.0)	24 (26.3)	0.26	−0.20
ACE-inhibitors and ARBs	81 (57.4)	215 (64.3)	0.16	0.16	52 (57.1)	54 (59.3)	0.76	0.05
Statins	102 (72.3)	280 (83.8)	0.004	0.38	68 (74.7)	67 (73.6)	0.87	−0.03
**Procedural profiles**								
Number of treated lesions	1.8 ± 1.1	1.7 ± 1.0	0.17	−0.14	1.7 ± 1.0	1.7 ± 1.1	0.89	−0.02
Number of treated vessels	1.3 ± 0.6	1.3 ± 0.5	0.35	−0.10	1.3 ± 0.5	1.3 ± 0.5	0.89	−0.02
**Treated vessels**								
Left main	7 (4.9)	10 (2.9)	0.29	−0.29	3 (3.2)	2 (2.1)	>0.99	−0.23
LAD	87 (61.7)	191 (57.1)	0.36	−0.10	53 (58.2)	52 (57.1)	0.88	−0.03
LCX	38 (26.9)	96 (28.7)	0.69	0.05	19 (20.8)	24 (26.3)	0.38	0.17
RCA	57 (40.4)	124 (37.1)	0.50	−0.08	40 (43.9)	36 (39.5)	0.55	−0.10
Type B2C lesions	132 (93.6)	315 (94.3)	0.77	0.07	87 (95.6)	84 (92.3)	0.54	−0.33
Multivessel disease	37 (26.2)	80 (23.9)	0.60	−0.07	21 (23.0)	22 (24.1)	0.86	0.03
Number of inserted stents	1.77 ± 0.98	1.65 ± 0.92	0.21	−0.13	1.67 ± 0.91	1.65 ± 0.95	0.87	−0.02
Average stent diameter (mm)	2.91 ± 0.37	3.00 ± 0.44	0.032	0.21	2.92 ± 0.39	2.96 ± 0.46	0.50	0.10
Total stent length (mm)	41.8 ± 26.7	39.1 ± 26.7	0.32	−0.10	39.8 ± 26.1	39.4 ± 25.8	0.90	−0.02
Bare metal stents	4 (2.8)	6 (1.7)	0.49	−0.26	2 (2.1)	2 (2.1)	>0.99	<0.001
Drug-eluting stents	139 (98.5)	329 (98.5)	0.95	−0.03	91 (100.0)	89 (97.8)	0.50	0.10
1st generation	29 (20.5)	64 (19.1)	0.72	−0.05	19 (20.8)	16 (17.5)	0.57	−0.12
2nd generation	110 (78.0)	265 (79.3)	0.75	0.04	72 (79.1)	73 (80.2)	0.85	0.04

Data are expressed as n (%), mean ± standard deviation (SD), or median (interquartile range). ACE = angiotensin converting enzyme; ARB = angiotensin receptor blocker; BMI = body mass index; CAD = coronary artery disease; CFS = Clinical Frailty Scale; CrCl = creatinine clearance; CT = computed tomography; hs-CRP = high sensitivity C-reactive protein; L1 = first lumbar vertebra; LDLc = low density lipoprotein cholesterol; LAD = left anterior descending artery; LCX = left circumflex artery; LVEF = left ventricular ejection fraction; PCI = percutaneous coronary intervention; PSM = propensity score-matched; RCA = right coronary artery; S.diff = standardized difference; SMA = skeletal muscle area; SMI = skeletal muscle index.

**Table 2 jcm-08-00712-t002:** Kaplan–Meier survival analysis of 3-year clinical outcomes.

	Overall Population			PSM Population		
	Low L1 SMI (*n =* 141)	High L1 SMI (*n =* 334)	Log-Rank *p*-Value	Low L1 SMI (*n =* 91)	High L1 SMI (*n =* 91)	Log-Rank *p*-Value
All-cause mortality	31 (23.7)	19 (5.9)	<0.001	22 (25.7)	9 (10.0)	0.009
Cardiovascular	10 (8.4)	7 (2.2)	0.004	7 (9.1)	5 (5.6)	0.47
Non-cardiovascular	21 (16.6)	12 (3.8)	<0.001	15 (18.3)	4 (4.7)	0.006
Non-fatal MI	11 (9.6)	6 (1.9)	<0.001	8 (10.4)	2 (2.4)	0.037
ST-elevation MI	6 (5.0)	3 (1.0)	0.008	4 (5.2)	1 (1.3)	0.14
Non-ST-elevation MI	5 (4.7)	3 (0.9)	0.022	4 (5.4)	1 (1.1)	0.14
Repeat revascularization	23 (25.0)	23 (7.9)	<0.001	16 (26.2)	8 (10.0)	0.025
TVR	18 (19.3)	18 (6.1)	<0.001	13 (21.1)	7 (8.9)	0.06
Non-TVR	8 (8.6)	7 (2.5)	0.011	6 (9.6)	1 (1.2)	0.032
MACE	49 (39.6)	37 (11.8)	<0.001	38 (46.1)	16 (18.1)	<0.001

Data are expressed as incidence (%). L1 = first lumbar vertebra; MACE = major adverse cardiovascular event; MI = myocardial infarction; PSM = propensity score-matched; SMI = skeletal muscle index; TVR = target vessel revascularization.

**Table 3 jcm-08-00712-t003:** Multivariate analysis of 3-year clinical outcomes.

	3-Year All-Cause Mortality		3-Year MACE	
	HR (95% CI)	*p*-Value	HR (95% CI)	*p*-Value
Model 1 *				
Low L1 SMI	4.34 (2.45–7.69)	<0.001	3.82 (2.54–5.74)	<0.001
Model 2 ^†^				
Low L1 SMI	3.74 (1.89–7.41)	<0.001	4.51 (2.76–7.38)	<0.001
Model 3 ^‡^				
Low L1 SMI	3.69 (1.81–7.52)	<0.001	3.77 (2.27–6.27)	<0.001
Model 4 ^§^				
Low L1 SMI	4.07 (1.95–8.45)	<0.001	3.76 (2.27–6.23)	<0.001
Model 5 ^||^ (PSM analysis)				
Low L1 SMI	2.68 (1.23–5.83)	0.013	2.80 (1.56–5.03)	0.001

* Univariate analysis; ^†^ adjusted for age, sex, and body mass index; ^‡^ further adjusted for left ventricular ejection fraction <50%, creatinine clearance <60 mL/min, Total cholesterol >5.17 mmol/L (>200 mg/dL), clopidogrel at discharge, statins at discharge, and average diameter of inserted stents; ^§^ further adjusted for clinical presentation as myocardial infarction, hypertension, diabetes, previous malignancy, multivessel disease, and implantation of second generation drug-eluting stents. ^||^ propensity score-matched analysis; further adjusted for dyslipidemia, number of treated lesions and treated vessels, number of implanted stents, and total stent length. CI = confidence interval; HR = hazard ratio; L1 = first lumbar vertebra; MACE = major adverse cardiovascular event; PSM = propensity score-matched; SMI = skeletal muscle index.

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
