# Peer review of "Prognostic Impact of Low Skeletal Muscle Mass on Major Adverse Cardiovascular Events in Coronary Artery Disease: A Propensity Score-Matched Analysis of a Single Center All-Comer Cohort"

_jcm, 2019, doi:10.3390/jcm8050712_

Reviewer 1 Report

In this prospective, single center study, Dr. Kang et al. demonstrated the association between sarcopenia assessed with CT and adverse clinical outcomes in patients with CAD undergoing PCI. They concluded that the CT-diagnosed sarcopenia was a strong predictor of adverse clinical outcomes in those patients. The following issues were noted during review;

 1.     The cut-off values of the SMA at the L1 level to predict 3-year mortality were used for the definitions of sarcopenia. As the author has mentioned in the manuscript, while the SMA at L3 level is widely accepted as a reference, the one at L1 level is not well validated. For the definition of sarcopenia, the cut-off values of SMA at L1 level should be obtained by DXA etc which reflect the whole-body skeletal muscle mass or frailty, not from the clinical outcomes. It might be better for the author to focus on identifying cut-off values of SMA at L1 level to predict all-cause mortality in patients undergoing PCI (not dividing them into sarcopenia and non-sarcopenia group).

2.     The follow-up rate was 82% in the present study. What were the characteristics of the patients who dropped-out? Were they more frail? 

3.     While sarcopenia was related to non-cardiac death, MI, and revascularization, no relation was observed between sarcopenia and cardiac death. Can an additional explanation be provided from the author?

4.     Cox-regression analysis was used for the analysis, but odds ratio was selected instead of hazard ratio.

5.     Higher incidence of myocardial infarction was observed in sarcopenia patients. Any information about the echocardiography during the follow-up period?

6.     It might be better to include the data of univariate analysis for the predictors of the adverse events.

 Author Response

The Reviewer #1’s comments and our responses

We appreciate for the reviewer’s constrictive comments. Thank you so much for your helpful feedback and these comments resulted in a substantial improvement of our manuscript.

 Point 1: The cut-off values of the SMA at the L1 level to predict 3-year mortality were used for the definitions of sarcopenia. As the author has mentioned in the manuscript, while the SMA at L3 level is widely accepted as a reference, the one at L1 level is not well validated. For the definition of sarcopenia, the cut-off values of SMA at L1 level should be obtained by DXA etc which reflect the whole-body skeletal muscle mass or frailty, not from the clinical outcomes. It might be better for the author to focus on identifying cut-off values of SMA at L1 level to predict all-cause mortality in patients undergoing PCI (not dividing them into sarcopenia and non-sarcopenia group).

 Response to Point 1:

The authors appreciate the reviewer’s kind comment about clarifying the definition of sarcopenia. The cut-off values of L1 SMI in this study were obtained based on the clinical outcomes, therefore, these values do not strictly satisfy the definition of sarcopenia. The main scope of the study was modified to focus on identifying the optimal cut-off values of L1 SMI to predict all-cause mortality in CAD patients undergoing PCI.

The name of previous study groups indicating the patients “with sarcopenia” and those “without sarcopenia (non-sarcopenia)” were changed into “low L1 SMI group” and “high L1 SMI group”, respectively. The term “sarcopenia” was mostly substituted by “low skeletal muscle mass (SMM)” in available cases. The title of the manuscript was also modified to reflect these changes. Please track the changes in the revised version of the main manuscript including abstract, tables and figures.

 Summary of Changes (Point 1):

The title of the manuscript was modified as followings: ‘Prognostic impact of low skeletal muscle mass on major adverse cardiovascular events in coronary artery disease: a propensity score-matched analysis of single center all-comer cohort’

The red colored text is the changes which has been rewritten and newly included into the introduction section of the manuscript: ‘This study investigated the long-term prognostic impact of CT-diagnosed low SMM in patients with CAD who underwent successful percutaneous coronary intervention (PCI). We identified the sex-specific cut-off values of SMM at the first lumbar (L1) vertebral level to predict all-cause mortality and validated its prognostic significance.’ (Page 2, line 57 - line 60)

Point 2: The follow-up rate was 82% in the present study. What were the characteristics of the patients who dropped-out? Were they more frail?

 Response to Point 2:

The authors compared the baseline characteristics and procedural profiles between the patients those who dropped-out (n=84) and those who completed the 3-year follow-up (n=391). Result of this additional analysis is provided in the Appendix of this response letter (Table A1). The patients who dropped-out were more likely to be presented as stable angina and showed frequent history of diabetes and peripheral artery disease. Laboratory results of high sensitivity C-reactive protein and glycated hemoglobin were higher and creatinine clearance was lower in the dropped-out patients than in those who completed the follow-up. Extent of CAD regarding the number of treated lesions, treated vessels and implanted stents was greater and the rate of multivessel disease was higher in the patients who completed the 3-year follow-up. The level of frailty assessed by the Canadian Study of Heath and Aging (CSHA) Clinical Frailty Scale (CFS) was similar in both groups.

** The result of this additional analysis is provided to the editor and reviewers only. This result is not included in the main manuscript. (check the attatched file)

 Point 3: While sarcopenia was related to non-cardiac death, MI, and revascularization, no relation was observed between sarcopenia and cardiac death. Can an additional explanation be provided from the author?

 Response to Point 3:

The authors appreciate the reviewer’s comment and this helped us to address the prognostic impact of low SMM more clearly. Additional explanations are included in the “paragraph 4.2.” of the discussion section about the predominant impact of low SMM on the non-cardiovascular death than the cardiovascular death. In brief, various medical comorbidities accompanied by low SMM are expected to have greater impact on death from non-cardiovascular cause than those from cardiovascular cause.

 Summary of Changes (Point 3):

The red colored text is the changes which has been rewritten and newly included into the introduction section of the manuscript: ‘Although the risk of cardiovascular death was attenuated in the propensity-score matched population, 1.6-fold increased risk of cardiovascular death still remained in the low L1 SMI group. The risk of non-cardiovascular death was significantly increased by over than 4-fold in the low L1 SMI group, suggesting that low SMM is likely to be accompanied by various medical comorbidities during long-term follow-up. Previous studies have reported that sarcopenia is closely related to the presence of medical comorbidities including chronic degenerative diseases [28-30]. These higher comorbidities associated with low SMM are expected to have greater impact on non-cardiovascular death than cardiovascular death and negatively affected the overall clinical outcomes in this study population.’ (Page 16, line 326 - line 333)

 Point 4: Cox-regression analysis was used for the analysis, but odds ratio was selected instead of hazard ratio.

Response to Point 4:

Thanks for your comments. The errors mentioned have been corrected with track changes (odds ratio [OR] à hazard ratio [HR]).

Point 5: Higher incidence of myocardial infarction was observed in sarcopenia patients. Any information about the echocardiography during the follow-up period?

 Response to Point 5:

During the 3-year follow-up, 270 (56.8%) of 475 patients and 118 (65.2%) of 182 patients were available of follow-up echocardiography results in the overall and propensity-score matched population, respectively. Follow-up left ventricular ejection fraction (LVEF) and changes from baseline LVEF were similar in both groups of L1 SMI. Results are provided in the Appendix of this response letter (Table A2).

** The result of this additional analysis is provided to the editor and reviewers only. This result is not included in the main manuscript (check the attatched file).

 Point 6: It might be better to include the data of univariate analysis for the predictors of the adverse events.

Response to Point 6:

   The data of univariate analysis for the predictors of the 3-year adverse outcomes are added to the Supplemental Table S3. (check the attached file)

  As a result, we feel the reviewers’ and the editors’ comments have resulted in a substantial improvement of our revised manuscript over the original one.

With my best regards,

Reviewer 2 Report

Journal of  Clinical  Medicine jcm-497069

Prognostic impact of sarcopenia on major adverse cardiovascular events in  coronary artery disease: a propensity score-matched analysis of single  center all-comer cohort

Dong Oh Kang et al.

Review

General comments

This is a very interesting study on the relevance of sarcopenia on the prognosis of patients with coronary artery disease (CAD). The Authors investigated the long-term prognostic impact of computed tomography (CT)-diagnosed sarcopenia in 475 patients who underwent successful PTCA. They found, in multivariate and propensity matched analysis, that sarcopenia was an independent predictor of all-cause mortality and major events at 3 years in patients with CAD undergoing PTCA.

The study is well designed and conducted, methods are clearly defined,  statistical analysis is correct and English language is satisfactory. The conclusion gives the important message that the prognosis of CAD patients is not only dependent from the classical cardiologic risk factors but also from general factors related to the general health condition of the patients.

Some major criticisms are however to be underlined:

Major points:

The presence of sarcopenia may be a marker of hidden confounders that have a negative influence on patients outcome, and that have not been reported in the paper, such as chronic degenerative or inflammatory diseases (i.e. chronic obstructive lung disease, cancer) and particularly frailty. This is  suggested by the over fourfold risk of non-cardiac death at 3 years in the propensity score analysis, compared to the less than 2 fold risk of cardiac death. In addition, included group had a significantly higher rate of non- cardiac mortality compared to excluded group (additional Table S4).  It is well known that sarcopenia is related to the presence of chronic disease that often are not clearly manifested but that may clearly develop in the years and lead to non- cardiac death. Although this concept is expressed in para 4.2, I believe that this should more explained and largely discussed. In addition, it would have been  important to know which were the causes of non-cardiac death in these patients. Can  the Authors provide these information?

In addition, the Authors should discuss the link between sarcopenia and frailty, conditions often associated in the elderly and leading to negative outcomes (Clegg A et al. Frailty in elderly people. Lancet 2013;381: 752e762). It is disappointing that  frailty was not assessed in these patients with any of the tools available in cardiovascular patients (Afilalo J et al.  Frailty Assessment in the Cardiovascular Care of Older Adults. JACC 2014; 63; 747–62). Frailty has been demonstrated to have an independent negative impact in the prognosis of CAD patients, included patients after PTCA (Singh M et al. Influence of Frailty and Health Status on Outcomes in Patients With Coronary Disease Undergoing Percutaneous Revascularization. Circ Cardiovasc Qual Outcomes. 2011;4:496-502).  It would have been  important to compare in this population the prognostic impact of frailty on the same end points assessed with CT sarcopenia. It is true that CT is the gold standard for sarcopenia definition, but in general practice this costly procedure with associated radiation burden may be not generally applicable. It is possible that common frailty tools, easily applicable, not invasive, may have given the same results of a more costly, risky and time taking instrument like CT. This is a possibility that should be considered and discussed.

Another hidden information  under sarcopenia presence could be the level of physical activity of these patients. These patients may be sarcopenic because of inactivity and sedentary habits, conditions associated with chronic disease, disability, poor psychosocial status and  negative cardiovascular or non CV events (Stenholm S et al. Association of Physical Activity History With Physical Function and Mortality in Old Age. J Gerontol A Biol Sci Med Sci, 2016; 71; 496–501 ). Do the Authors have information on any of these conditions?

In addition, since the Authors used the cross-sectional area of skeletal  muscle at the first lumbar (L1) vertebral level as definition of sarcopenia  (<31.00 cm2/m2 for male; <25.00 cm2/m2 for female), did they try to see if their result would have changed by adopting a different sarcopenia definition cut point?

Minor points:

When citing CR as a class I indications for PTCA pts, 2016 European Guidelines on cardiovascular disease prevention in clinical practice should be also cited.

 Author Response

The Reviewer #2’s comments and our responses

We appreciate for the reviewer’s constrictive comments. Thank you so much for your helpful feedback and these comments resulted in a substantial improvement of our manuscript.

 # Major Point 1: The presence of sarcopenia may be a marker of hidden confounders that have a negative influence on patient’s outcome, and that have not been reported in the paper, such as chronic degenerative or inflammatory diseases (i.e. chronic obstructive lung disease, cancer) and particularly frailty. This is suggested by the over fourfold risk of non-cardiac death at 3 years in the propensity score analysis, compared to the less than 2 fold risk of cardiac death. In addition, included group had a significantly higher rate of non- cardiac mortality compared to the excluded group (additional Table S4). It is well known that sarcopenia is related to the presence of chronic disease that often are not clearly manifested but that may clearly develop in the years and lead to non- cardiac death. Although this concept is expressed in para 4.2, I believe that this should more explained and largely discussed. In addition, it would have been important to know which were the causes of non-cardiac death in these patients. Can the Authors provide these information?

 Response to Major Point 1:

1-1) Relationship between sarcopenia and medical comorbidities

The authors strongly agree with the reviewer’s comment about emphasizing the relationship between sarcopenia and accompanying medical comorbidities which could lead to non-cardiovascular death in the future. Since the presence of sarcopenia (low skeletal muscle mass; SMM) could be a marker of hidden confounders, incidental detection of sarcopenia could provide a chance to discover a concealed medical comorbidities. Therefore, the authors have incorporated this concept into the “paragraph 4.2.” in the discussion section with additional explanation and recommendation. We also modified the subtitle of “paragraph 4.2.” to emphasize this relationship between sarcopenia and accompanying medical comorbidities.

 Summary of Changes (Major Point 1-1):

   The subtitle of paragraph 4.2 is rewritten as followings: ‘4.2. Assessment of SMM by L1 SMI as a promising marker of prognosis and comorbidity’ (Page 16, line 320)

 The red colored text is the changes which has been newly included into the discussion section of the manuscript: ‘Previous studies have reported that sarcopenia is closely related to the presence of medical comorbidities including chronic degenerative diseases [28-30]. These higher comorbidities associated with low SMM are expected to have greater impact on non-cardiovascular death than cardiovascular death and negatively affected the overall clinical outcomes in this study population. Incidental detection of low SMM could provide a chance to discover a concealed chronic illness often before its clinical manifestation. The findings of the present study suggests that CAD patients with low SMM should consider undergoing comprehensive investigation to discover accompanying medical comorbidities.’ (Page 16, line 329 - line 335)

 1-2) Specific cause of non-cardiovascular death

Specific causes of non-cardiovascular death were classified into five categories: a) infection, b) fatal bleeding due to intracranial hemorrhage and gastrointestinal bleeding, c) respiratory disease, d) renal failure, and e) other causes. Death from infectious disease was the most common cause of non-cardiovascular death, and detailed information is included in the Supplemental Data as Table S2.

 Summary of Changes (Major Point 1-2):

The red colored text is the changes which has been newly included into results section of the manuscript: ‘Detailed information about the cause of non-cardiovascular death is provided in Table S2.’ (Page 7, line 208 - line 209)

Table S2 --> Please check the attatched file.

# Major Point 2: In addition, the Authors should discuss the link between sarcopenia and frailty, conditions often associated in the elderly and leading to negative outcomes (Clegg A et al. Frailty in elderly people. Lancet 2013;381: 752e762). It is disappointing that frailty was not assessed in these patients with any of the tools available in cardiovascular patients (Afilalo J et al.  Frailty Assessment in the Cardiovascular Care of Older Adults. JACC 2014; 63; 747–62). Frailty has been demonstrated to have an independent negative impact in the prognosis of CAD patients, included patients after PTCA (Singh M et al. Influence of Frailty and Health Status on Outcomes in Patients With Coronary Disease Undergoing Percutaneous Revascularization. Circ Cardiovasc Qual Outcomes. 2011;4:496-502).  It would have been important to compare in this population the prognostic impact of frailty on the same end points assessed with CT sarcopenia. It is true that CT is the gold standard for sarcopenia definition, but in general practice this costly procedure with associated radiation burden may be not generally applicable. It is possible that common frailty tools, easily applicable, not invasive, may have given the same results of a more costly, risky and time taking instrument like CT. This is a possibility that should be considered and discussed.

Another hidden information under sarcopenia presence could be the level of physical activity of these patients. These patients may be sarcopenic because of inactivity and sedentary habits, conditions associated with chronic disease, disability, poor psychosocial status and negative cardiovascular or non CV events (Stenholm S et al. Association of Physical Activity History With Physical Function and Mortality in Old Age. J Gerontol A Biol Sci Med Sci, 2016; 71; 496–501). Do the Authors have information on any of these conditions?

 Response to Major Point 2:

2-1) Additional assessment of frailty and physical activity

The authors would like to appreciate the reviewer’s comment by pointing out this important relationship between sarcopenia and frailty, and providing us a valuable opportunity to improve our manuscript.

From 2004 to 2014, KUGH PCI registry did not routinely included the results of exercise functional assessment such as hand-grip strength test, 6-minute walking test or maximum oxygen consumption (VO2max) test. During the study period, instead of these exercise assessments, detailed investigation on the patient’s activity of daily living (ADL) and level of functional independence was undertaken by the attending physician at the time of index admission. The patient’s level of frailty was assessed by using the Canadian Study of Health and Aging (CSHA) Clinical Frailty Scale (CFS) [1]. This CSHA CFS consists of a 7-point scale and it has been validated in patients with CAD, especially in those with NSTEMI [2]. This CFS enables the integrated assessment of frailty and physical activity, and also has an advantage of easy accessibility in a real-world clinical practice. We defined the presence of frailty by CFS 5 and the patients were divided into two groups: non-frail group (CFS<5; n=354) and frail group (CFS 5; n=121). Frail group was further divided into mild to moderate frailty (CFS 5-6; n=105) and severe frailty (CFS 7; n=16).

The level of frailty assessed by CFS is included in the tables presenting baseline characteristics (Table 1, Supplemental Table S6). As recommended by the reviewer, the authors investigated the prognostic impact of frailty in this study population, and the results of adverse clinical outcomes and multiple stepwise Cox regression analysis are provided in Supplemental Table S4 and S5, respectively. In brief, the low L1 SMI group showed a greater level of frailty than the high L1 SMI group. Frail patients showed significantly higher incidence of 3-year all-cause mortality and MACEs than the non-frail patients. In multivariate analysis, the presence of frailty was an independent predictor of 3-year all-cause mortality.

 Summary of Changes (Major Point 2-1):

The red colored text is the changes which has been newly included into the methods section of the manuscript:

The Canadian Study of Health and Aging (CSHA) Clinical Frailty Scale (CFS) [19] was used to measure the level of frailty prior to admission. The presence of frailty was defined by CSHA CFS ≥5; mild to moderate (CFS 5-6) and severe (CFS 7) frailty.’ (Page 3, line 90 - line 93)

 The red colored text is the changes which has been newly included into the results section of the manuscript:
3.6. Clinical outcomes based on the presence of frailty

Detailed study results based on the presence of frailty are provided in Table S4 and Table S5. Frail patients showed significantly higher incidence of 3-year all-cause mortality (24.8% vs. 6.4%; HR: 3.99; 95% CI: 2.28 to 6.98; p<0.001) and MACEs (33.4% vs. 17.8%; HR: 2.07; 95% CI: 1.37 to 3.13; p <0.001) than non-frail patients. In fully adjusted multivariate analysis, the presence of frailty was an independent predictor of 3-year all-cause mortality (HR: 2.81; 95% CI: 1.47 to 5.39; p=0.002), however, the impact of frailty on 3-year MACEs was markedly attenuated (HR: 1.29; 95% CI: 0.81 to 2.05; p=0.28).’ (Page 15, line 263 - line 270)

Table S4 and Table S5 --> Please check the attatched file

 2-2) Comparison of CT-defined L1 SMI with level of frailty and future perspectives

The authors generally agree with the reviewer’s comment about using a more convenient frailty assessment tool in a general practice. In a real-world practice, frailty assessment with simple exercise test or skeletal muscle mass measurement with other simple methods like BIA or DEXA could be more feasible. Further studies are required in this population to validate the prognostic impact of sarcopenia by using alternative techniques, such as DEXA or BIA, and comparison with frailty scales should be conducted.

CT scan still remain as the gold standard for diagnosing sarcopenia and could provide more sophisticated diagnostic and prognostic information compared with the simple frailty assessment as shown in the present study results. Recently, evaluation of CAD with coronary CT is rapidly increasing, therefore, CT images to assess skeletal muscle mass at the L1 level could be easily acquired without giving additional harm to the patients.

The authors elaborated the discussion section by reflecting the reviewer’s suggestions and refined the contents as described above. References provided by the reviewer were thoroughly reviewed by the authors and two of the references are cited in the revised version. Thank you once again for providing a valuable information to improve our manuscript.
: (Singh M et al. Circ Cardiovasc Qual Outcomes. 2011;4:496-502. >
as reference 31)
: (Clegg A et al. Lancet 2013;381: 752-762. > as reference 32)

 Summary of Changes (Major Point 2-2):

The red colored text is the changes which has been newly included into the discussion section of the manuscript: ‘In the present study, patients with low L1 SMI and frailty showed a similar pattern on the risk of 3-year adverse outcomes. Negative impact of frailty on patients with CAD undergoing PCI was consistent with the previous study results [31]. It is well known that both sarcopenia and frailty shares an overlapping characteristics in a variety of pathophysiologic conditions [32-34]. The CFS was able to predict the risk of 3-year all-cause mortality similar to the CT-determined low L1 SMI, however, the prediction of other patient-oriented outcomes were limited. Although CT scans are expensive and associated with radiation burden, detailed characterization of muscle loss is able to provide better diagnostic and prognostic information. Further studies to investigate the prognostic impact of sarcopenia assessed by alternative methods, such as BIA or DEXA, should be conducted in this population, and comparison with frailty scales should be performed to provide additional evidences.’ (Page 16, line 337 - line 346)

The red colored text is the changes which has been newly included into the discussion section of the manuscript: ‘Under the widespread use of coronary CT in CAD, CT images to assess SMM at the L1 level could be easily acquired without additional radiation burden or extra payment in this population. ’ (Page 16, line 374 - line 376)

 # Major Point 3: In addition, since the Authors used the cross-sectional area of skeletal  muscle at the first lumbar (L1) vertebral level as definition of sarcopenia (<31.00 cm2/m2 for male; <25.00 cm2/m2 for female), did they try to see if their result would have changed by adopting a different sarcopenia definition cut point?

 Response to Major Point 3:

The authors have performed an additional analysis by adopting a different cut-off values of L1 SMI to define sarcopenia which were provided by Derstine et al [3] (<34.60 cm2/m2 for male;<25.90 cm2/m2 for female). The 3-year clinical outcomes from the additional cut-point analysis are provided in the Appendix of this response letter (Table A3 and Figure A1).

Brief results: Incidence of 3-year all-cause mortality and MACEs were significantly higher in the patients with sarcopenia than in those without sarcopenia. The propensity-score matched results of the additional analysis were similar to the results of the overall population. The result of adopting a different cut-off value to define sarcopenia was consistent to the findings of the original analysis. The cut-off values of the additional analysis are originally derived from the healthy North American population, therefore, these values are exploratory to our study population.

**The results of the additional analysis are provided exclusively to the editor and reviewers only. These results are not included in the main manuscript.

Table A3 and Figure A1 --> Please check the attatched file.

# Minor Point 1: When citing CR as a class I indications for PTCA pts, 2016 European Guidelines on cardiovascular disease prevention in clinical practice should be also cited.

 Response to Minor Point 1:

The authors included the recommended article as a reference to support CR as a class I indication for CAD patients undergoing PCI. This article is cited as reference number 38.

 Summary of Changes (Minor Point 1):

The red colored text is the changes which has been newly included into the reference section of the manuscript:

38. Piepoli, M.F.; Hoes, A.W.; Agewall, S.; Albus, C.; Brotons, C.; Catapano, A.L.; Cooney, M.T.; Corra, U.; Cosyns, B.; Deaton, C., et al. 2016 European Guidelines on cardiovascular disease prevention in clinical practice: The Sixth Joint Task Force of the European Society of Cardiology and Other Societies on Cardiovascular Disease Prevention in Clinical Practice (constituted by representatives of 10 societies and by invited experts)Developed with the special contribution of the European Association for Cardiovascular Prevention & Rehabilitation (EACPR). European heart journal 2016, 37, 2315-2381, doi:10.1093/eurheartj/ehw106.’ (Page 23, line 561 - line 567)

As a result, we feel the reviewers’ and the editors’ comments have resulted in a substantial improvement of our revised manuscript over the original one.

With my best regards,

References

1.        Rockwood, K.; Song, X.; MacKnight, C.; Bergman, H.; Hogan, D.B.; McDowell, I.; Mitnitski, A. A global clinical measure of fitness and frailty in elderly people. CMAJ : Canadian Medical Association journal = journal de l'Association medicale canadienne 2005, 173, 489-495, doi:10.1503/cmaj.050051.

2.        Ekerstad, N.; Swahn, E.; Janzon, M.; Alfredsson, J.; Lofmark, R.; Lindenberger, M.; Carlsson, P. Frailty is independently associated with short-term outcomes for elderly patients with non-ST-segment elevation myocardial infarction. Circulation 2011, 124, 2397-2404, doi:10.1161/circulationaha.111.025452.

3.        Derstine, B.A.; Holcombe, S.A.; Ross, B.E.; Wang, N.C.; Su, G.L.; Wang, S.C. Skeletal muscle cutoff values for sarcopenia diagnosis using T10 to L5 measurements in a healthy US population. Scientific reports 2018, 8, 11369, doi:10.1038/s41598-018-29825-5.

Round  2

Reviewer 1 Report

The authors corrected the manuscript enough according to the comments. The revised manuscript is suitable for publication.

Reviewer 2 Report

The Authors have adequately replied to all points raised and made the appropriate changes in manuscript and tables.